# Long non-coding RNA *NR2F1-AS1* induces breast cancer lung metastatic dormancy by regulating NR2F1 and ΔNp63

Yingjie Liu[1,2,7], Peiyuan Zhang[1,7], Qiuyao Wu[1], Houqin Fang[3], Yuan Wang[1], Yansen Xiao[1], Min Cong[1], Tingting Wang[3], Yunfei He[1], Chengxin Ma[1], Pu Tian[1], Yajun Liang[1], Lun-Xiu Qin[4], Qingcheng Yang[5], Qifeng Yang [6], Lujian Liao[3] & Guohong Hu [1,2✉]

Disseminated tumor cells often fall into a long term of dormant stage, characterized by decreased proliferation but sustained survival, in distant organs before awakening for metastatic growth. However, the regulatory mechanism of metastatic dormancy and awakening is largely unknown. Here, we show that the epithelial-like and mesenchymal-like subpopulations of breast cancer stem-like cells (BCSCs) demonstrate different levels of dormancy and tumorigenicity in lungs. The long non-coding RNA (lncRNA) *NR2F1-AS1* (*NAS1*) is up-regulated in the dormant mesenchymal-like BCSCs, and functionally promotes tumor dissemination but reduces proliferation in lungs. Mechanistically, *NAS1* binds to *NR2F1* mRNA and recruits the RNA-binding protein PTBP1 to promote internal ribosome entry site (IRES)-mediated NR2F1 translation, thus leading to suppression of *ΔNp63* transcription by NR2F1. Furthermore, ΔNp63 downregulatio results in epithelial-mesenchymal transition, reduced tumorigenicity and enhanced dormancy of cancer cells in lungs. Overall, the study links BCSC plasticity with metastatic dormancy, and reveals the lncRNA as an important regulator of both processes.

[1] CAS Key Laboratory of Tissue Microenvironment and Tumor, Shanghai Institute of Nutrition and Health, University of Chinese Academy of Sciences, Chinese Academy of Sciences, Shanghai, China. [2] Shanghai Institute of Nutrition and Health, Shanghai Jiao Tong University School of Medicine (SJTUSM) & Chinese Academy of Sciences, Shanghai, China. [3] Shanghai Key Laboratory of Regulatory Biology, School of Life Sciences, East China Normal University, Shanghai, China. [4] Department of General Surgery, Huashan Hospital, Fudan University, Shanghai, China. [5] Department of Orthopedics, Shanghai Jiao Tong University Affiliated Sixth People's Hospital, Shanghai, China. [6] Department of Breast Surgery, Qilu Hospital of Shandong University, Ji'nan, China. [7] These authors contributed equally: Yingjie Liu, Peiyuan Zhang. ✉email: ghhu@sibs.ac.cn

Breast cancer is a major threat to women's health, and metastasis of tumor cells into vital organs such as lung, brain, liver, and bone accounts for most deaths of breast cancer patients. During the process of metastasis, disseminated tumor cells (DTCs) often fall into a long-term of dormant stage, characterized by slow proliferation, sustained survival, and resistance to adjuvant chemotherapy, in distant organs[1–3]. Metastatic dormancy represents a menace in the clinic as solitary tumor cells or foci can exist for months, years, or even decades without being detected while with the possibility to reactivate and lead to tumor recurrence[4,5]. The clinical importance of tumor dormancy has long been recognized[6–10]. A number of elegant studies have revealed some tumor-intrinsic factors, including p38, TGFβ, and NF2F1, which regulate the dormant state of cancer cells[11–17]. The tumor microenvironment has also been shown to influence the dormancy and reactivation of disseminated cancer cells[15–20]. However, the mechanisms regulating metastatic dormancy are far from being completely understood. In particular, it is intriguing why the tumor cells fall into proliferation arrest after aggressive growth and spreading at the primary sites, and what causes their reactivation thereafter.

Previous studies have noticed that dormant tumor cells might recapitulate normal stem cells for the themes of regulation of growth arrest and prolonged pluripotency[4,21]. Recently the parallels of dormant DTCs to cancer stem-like cells (CSCs) have also been discussed[22,23]. More importantly, dormant tumor cells retain the capacity to initiate new tumors, a defining feature of CSCs, after a long term of quiescence. It has been reported that latent metastatic cells of breast cancer were enriched for the CD44$^{high}$CD24$^{low}$ population[20], a typical profile of CSCs in breast cancer. However, it was also observed that the stromal bone morphogenetic protein (BMP) produced by lung resident cells induces the dormancy of DTCs via impairing their stemness, while the BMP inhibitor COCO enhances the stem-like traits of latent cells and reactivates these cells[12]. Additional studies also showed that the acquisition of stem cell properties is crucial for the colonization and outgrowth of DTCs[24,25]. Therefore, the current understanding of the relationship between metastatic dormancy and cancer stemness is seemingly inconsistent, and additional work is necessary to elucidate the role of CSCs in metastatic dormancy and reactivation.

In fact, CSCs, which refer to the minor subpopulation of heterogeneous tumor cells with enhanced tumor-initiating ability, could be heterogeneous in themselves. In breast cancer, CSCs were initially identified by low or negative expression of CD24 and high expression of CD44[26]. Subsequent studies showed that high aldehyde dehydrogenase (ALDH) activity also marks breast cancer stem-like cells (BCSCs)[27]. CD24$^-$CD44$^+$ BCSCs and ALDH$^+$ BCSCs are largely distinct populations with mesenchymal and epithelial characteristics, respectively[28]. Interestingly, functional studies showed that ALDH$^+$ BCSCs are more proliferative and tumorigenic than CD24$^-$CD44$^+$ BCSCs, while ALDH$^-$CD24$^-$CD44$^+$ mesenchymal BCSCs tend to be quiescent[27,28]. This is in line with the notion that cancer cells after epithelial–mesenchymal transition (EMT) tend to be growth-arrested[29,30]. These studies could indicate that CSCs heterogeneity might offer an explanation to the inconsistent observations regarding the roles of CSCs in metastatic dormancy.

Recently, lncRNAs have been found to regulate various cancer processes[31–33], such as EMT, proliferation, survival, and metastasis[34–36]. The expression of a few lncRNAs has been also linked to metastatic dormancy. Gooding et al. reported the differential expression of the lncRNA BORG in dormant breast cancer cell lines[37,38]. In addition, NAS1 was also found to be upregulated in the late relapse of estrogen receptor-positive (ER$^+$) breast cancer and could regulate in vitro cell growth[39]. It was also reported to regulate the progression of several other cancer types through its microRNA-sponging function[40–44]. However, the functional roles of these lncRNAs in metastatic dormancy have not been validated.

In this study, we show that lncRNA NAS1 is highly expressed in mesenchymal-like BCSCs and reveal its functions to promote cancer cell dissemination and metastatic dormancy in the lungs by regulating the NR2F1-ΔNp63 axis.

## Results

**The mesenchymal subpopulation of BCSCs is prone to metastatic dormancy in the lungs.** A number of studies have reported the heterogeneity of BCSCs. Previously we also found two CSC subpopulations in breast cancer with different CD44 staining intensities and metastatic capacities[45]. By CD44 and CD24 flow cytometry, a CD24$^-$CD44$^{med}$ subpopulation (referred to as P1 thereafter) with positive but weak CD44 staining, and a CD24$^-$CD44$^{high}$ subpopulation (referred to as P2 thereafter) with stronger CD44 staining were observed in breast tumors and the isogenic MCF10 breast cancer cell line series that include MCF10AT, MCF10CA1h, and MCF10CA1a[46]. Both P1 and P2 subpopulations demonstrated enhanced tumorigenic capacity as compared to the non-CSC cells, but only P1 tumors resulted in lung metastases in mice[45]. Interestingly, P1 and the metastatic cell line MCF10CA1a that is enriched with P1 cells (Supplementary Fig. 1a) demonstrated epithelial morphology and expressed the epithelial marker CDH1, while P2 was mesenchymal-like with enhanced in vitro migratory and invasive properties (Supplementary Fig. 1b–e). Partial conversion from P1 to P2 cells was observed after multiple passages in serum-containing culture media, but not in serum-free media. When the cells were orthotopically injected into mice, P2 tumors also displayed higher invasive capacity (Fig. 1a) and resulted in more circulating cancer cells than P1 tumors in the blood of mice (Supplementary Fig. 1f). The less metastatic cell line MCF10CA1h, containing both P1 and P2 cells, displayed mesenchymal appearance in the majority but with occasional epithelial cell clusters (Supplementary Fig. 1a, b). In addition, we found that P1 contained a significantly higher ALDH$^+$ fraction as compared to P2 (Supplementary Fig. 1g). MCF10CA1a also displayed higher ALDH activity than MCF10CA1h (Supplementary Fig. 1g, h). These observations were in line with the previous study[28] reporting that ALDH and CD44 mark epithelial-like and mesenchymal-like, respectively, subsets of BCSCs. Furthermore, we performed RNA-sequencing profiling of MCF10CA1h and its P1 and P2 subpopulations (named as CA1h-P1 and CA1h-P2 thereafter), as well as MCF10CA1a, followed by gene set enrichment analysis (GSEA). The analyses showed that the previously reported epithelial-like BCSC (E-BCSC) signature and mesenchymal-like BCSC (M-BCSC) signature[28] were enriched in the metastatic (MCF10CA1a and CA1h-P1) and less metastatic (MCF10CA1h and CA1h-P2) cells, respectively (Supplementary Fig. 1i).

Since mesenchymal-like P2 cells display higher capacities of migration, invasion, and dissemination but are less lung-metastatic than the epithelial-like P1 cells[45], we sought to analyze the post-dissemination process of metastasis for these cells. MCF10CA1a and MCF10CA1h, as well as CA1h-P1 and CA1h-P2, were labeled with firefly luciferase and GFP, and intravenously inoculated into athymic mice for analysis of lung metastasis. The proliferation of tumor cells seeded in the lungs was analyzed by 5-ethynyl-2'-deoxyuridine (EdU) labeling 24 h prior to lung harvest. It was observed that more GFP$^+$ MCF10CA1h and CA1h-P2 cells were seeded in lungs compared with MCF10CA1a and CA1h-P1, respectively, after intravenous inoculation (Fig. 1b, c). However, lower percentages of

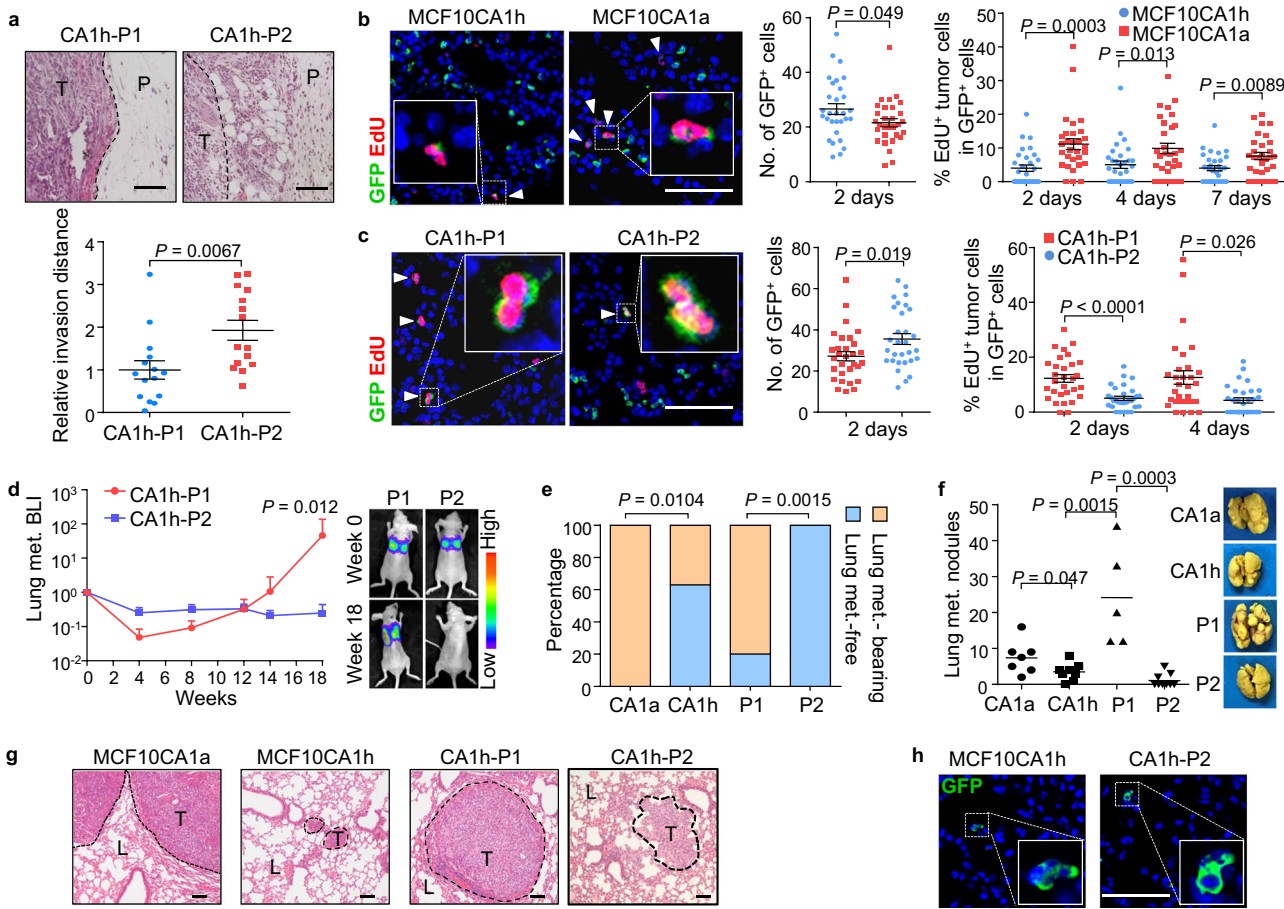

**Fig. 1 The P2 BCSC subpopulation displays a dormant phenotype in the lungs. a** H&E staining (top) and distance from tumor edges to invasive fronts (bottom) after orthotopic injection of CA1h-P1 and CA1h-P2 cells ($n = 15$ random microscopic fields (RMFs) from 5 tumors each group). **b, c** Seeding and proliferation of MCF10CA1h and MCF10CA1a (**b**), or MCF10CA1h subpopulations (**c**) in lungs at 2–7 days after intravenous transplantation Shown are representative immunofluorescences (IF) images (left), a number of GFP⁺ tumor cells (middle) and EdU⁺ proportions of tumor cells (right); $n$ = RMFs from 3 mice. The triangles indicate GFP and EdU double-positive cells. **d–h** Long-term analyses of lung metastasis after intravenous transplantation of cancer cells ($n = 7, 8, 5$, and 9 mice for MCF10CA1a, MCF10CA1h, CA1h-P1, and CA1h-P2, respectively). Shown are BLI quantification of pulmonary metastasis by CA1h-P1 and CA1h-P2 (**d**), percentages of mice with or without metastatic signal in lungs at week 14 (**e**), quantitation (top) and representative images (bottom) of pulmonary surface nodules (**f**), H&E (**g**), and IF (**h**) staining of lung tissue sections at week 20. H&E and IF staining in **g, h** were performed on lung sections of three mice in each group with similar results and representative images were shown. In **a, g**: L lung, P para-tumor, T tumor; dashed lines indicate tumor edges. Scale bar, 100 μm. Data represent mean ± SEM (**a–c**), mean ± SD (**d**), or mean with data points (**f**). Statistical significance was determined by a two-tailed unpaired $t$ test (**a–c, f**), two-sided chi-square test (**e**), or two-sided Mann–Whitney test (**d**).

MCF10CA1h and CA1h-P2 cells were active in proliferation at different time points of the early stage of colonization (Fig. 1b, c). Concordantly, in vitro analysis showed that although these cells only displayed minor differences of cell cycle distribution in nutrition-sufficient culture (Supplementary Fig. 2a), significantly more CA1h-P2 cells were arrested in G0/G1 phases than CA1h-P1 in serum-free culture (Supplementary Fig. 2b). Long-term metastasis monitoring by bioluminescent imaging (BLI) of the mice also showed the steady growth of CA1h-P1 tumors in lungs after the initial seeding phase, while the signals of CA1h-P2 cells remained almost unchanged after 18 weeks albeit with a stronger initial intensity (Fig. 1d). Significantly more mice inoculated with MCF10CA1a and CA1h-P1 cells developed lung metastasis in the end (Fig. 1e), with apparent tumor nodules in the lung surface (Fig. 1f) and aggressive tumor areas in the lung sections (Fig. 1g), while MCF10CA1h and CA1h-P2 remained as single cells or small foci for 20 weeks (Fig. 1g, h), and led to much fewer tumor nodules (Fig. 1f). These results showed that MCF10CA1h and CA1h-P2, rather than MCFCA1a or CA1h-P1, display a dormant phenotype in the lungs.

**NR2F1-AS1 is upregulated in latent cells and promotes metastatic dormancy**. Then we compared the RNA sequencing profiles between the lung-dormant cells (MCF10CA1h and CA1h-P2) and the lung-metastatic cells (MCF10CA1a and CA1h-P1), as well as MCF10AT, which forms the only carcinoma in situ and cannot disseminate to lungs in mice[46]. We primarily focused on lncRNAs whose roles were barely explored in metastatic dormancy and identified 18 upregulated and 7 downregulated lncRNAs in the lung-dormant cells (Fig. 2a). Notably, *NR2F1-AS1* (termed as *NAS1* thereafter), located upstream of the protein-encoding gene *NR2F1* but transcribed in the opposite direction, was among the upregulated lncRNAs. In addition, analysis of a triple-negative breast cancer clinical cohort[47] showed that *NAS1* was upregulated in late-recurring tumors (relapsed after 2 years following diagnosis) than the tumors that relapsed within the first 2 years after diagnosis (Supplementary Fig. 3a). The latest study by Sanchez Calle et al.[39] also reported the correlation of *NAS1* expression to late recurrence of ER⁺ breast cancer. Re-analysis of the ER⁺ cohort in this study confirmed the upregulation of *NAS1* in the late relapse of breast cancer (Supplementary Fig. 3b).

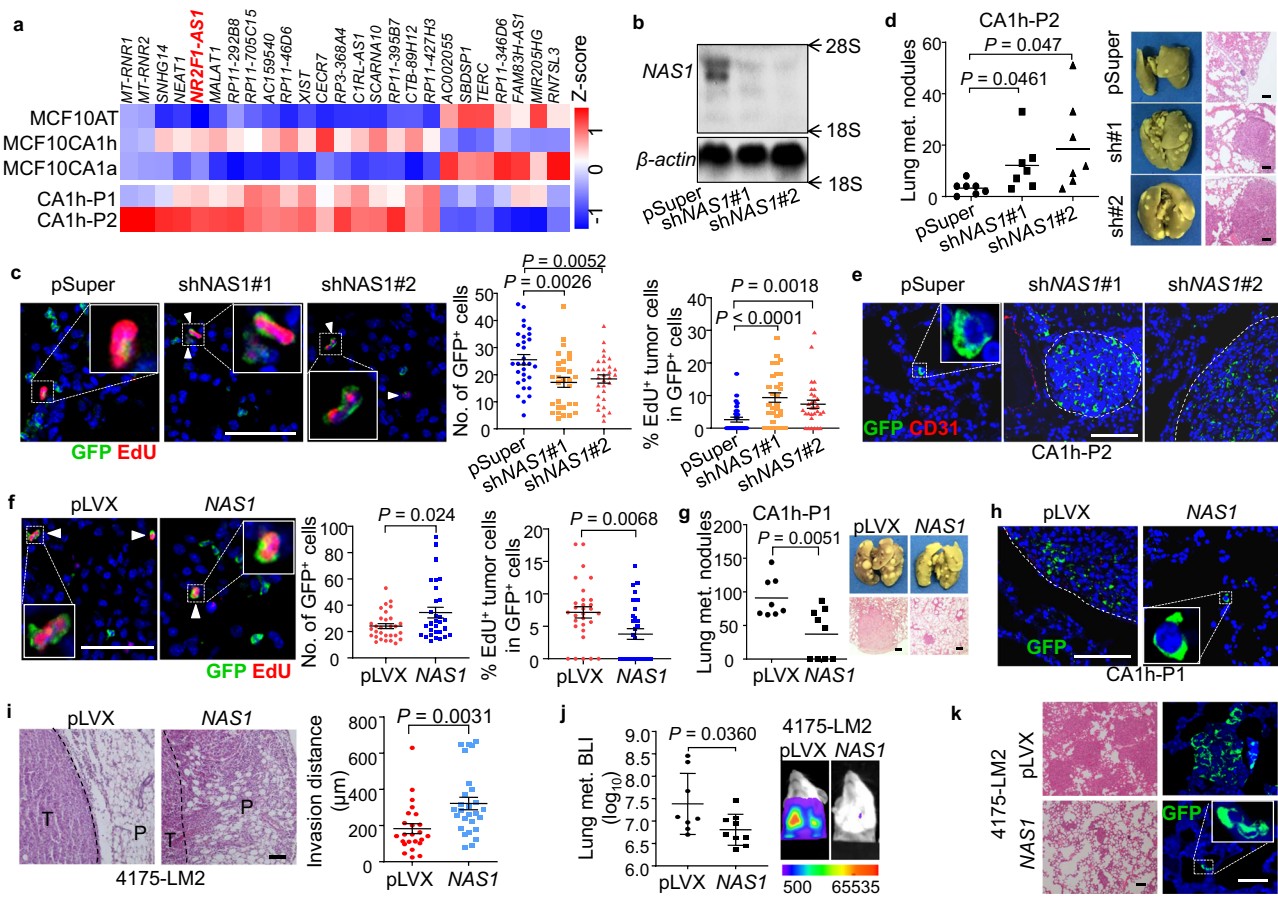

**Fig. 2 NR2F1-AS1 (NAS1) promotes metastatic dormancy of breast cancer. a** Expression heatmap of differential lncRNAs in MCF10 cell lines. **b** Northern blot of NAS1 in CA1h-P2 after NAS1 knockdown, with 18S and 28S ribosomal RNAs as the markers. Northern blotting was repeated three times independently with similar results. **c** Analyses of total and proliferative GFP$^+$ tumor cells seeded in lungs 4 days after intravenous injection of CA1h-P2 with NAS1 knockdown (n = 30 RMFs from 3 mice of each group). The triangles indicate GFP and EdU double-positive cells. **d** Numbers of pulmonary nodules 4 months after intravenous injection of CA1h-P2 with NAS1 knockdown (n = 7 mice in each group). **e** IF staining of lung tissue sections in (**d**). **f** Analyses of total and proliferative GFP+ tumor cells seeded in lungs 4 days after intravenous injection of CA1h-P1 with NAS1 overexpression (n = 30 RMFs from 3 mice). The triangles indicate GFP and EdU double-positive cells. **g** Number of pulmonary nodules 4 months after intravenous injection of CA1h-P1 with NAS1 overexpression (n = 8 and 9 mice without or with NAS1 overexpression, respectively). **h** IF staining of lung tissue sections in (**g**). **i** H&E staining (left) and invasion depth (right) of tumor edges after orthotopic injection of NAS1-overexpressing 4175-LM2 cells in NOD/SCID mice (n = 24 and 27 RMFs from 8 and 9 mice, respectively). **j** BLI quantification (left) and representative images (right) of lung metastasis of the mice in (**i**) 9 weeks after injection (n = 8 and 9 mice without or with NAS1 overexpression, respectively). **k** H&E and IF staining of lung tissue sections in (**j**). H&E and IF staining in **e**, **h**, **k** were performed on lung sections of three mice in each group with similar results, and representative images were shown. Scale bar, 100 μm. Data represent mean ± SD (**j**), mean ± SEM (**c**, **f**, **I**), or mean with data points (**d**, **g**). Statistical significance was determined by a two-tailed unpaired t test (**c**, **d**, **f**, **g**, **i**) or two-sided Mann–Whitney test (**j**).

Furthermore, fluorescence in situ hybridization analyses of NAS1 showed NAS1 was expressed in primary tumors and small tumor cell clusters in the lungs of MCF10CA1h, but not in the large lung metastatic nodules of the cells (Supplementary Fig. 3c). These results indicated an association of NAS1 expression with relapse latency.

Bioinformatic analyses by PhyloCSF[48] and Coding Potential Calculator[49] both confirmed the lack of coding potential of NAS1 (Supplementary Fig. 4a, b). Therefore, we performed 5′ and 3′ rapid amplification of cDNA ends (RACE) assays of NAS1 (Supplementary Fig. 4c) and discovered its full length of 2956 bp (Supplementary Table 1), which was consistent with that shown by Northern blotting of NAS1 (Fig. 2b). We also analyzed the subcellular distribution of NAS1 and observed uniform expression in both the cytoplasm and nucleus of the cancer cells (Supplementary Fig. 4d). Importantly, quantitative polymerase chain reaction (qPCR) analyses confirmed the elevated expression

of NAS1 in lung-dormant MCF10CA1h and CA1h-P2 cells (Supplementary Fig. 4e, f).

To study the role of NAS1 in metastatic dormancy, we knocked down NAS1 with short hairpin RNAs (shRNAs) in CA1h-P2 (Fig. 2b and Supplementary Fig. 4g). NAS1 knockdown resulted in transcriptomic changes of CA1h-P2 toward the P1-like state as shown by GSEA analysis (Supplementary Fig. 5a), and promoted the proliferation of CA1h-P2 cells under nutrient stress (Supplementary Fig. 5b). Importantly, when the cells were intravenously inoculated into nude mice, we observed fewer GFP$^+$ cancer cells initially seeded in the lungs after NAS1 knockdown, but these cells were more proliferative (Fig. 2c), leading to enhanced metastatic outgrowth and significantly more metastatic nodules, instead of solitary tumor cells, in lungs after 4 months (Fig. 2d, e). In contrast, NAS1 overexpression in MCF10AT and CA1h-P1 (Supplementary Fig. 4g) enhanced initial cancer cell seeding in lungs, but reduced the proliferation

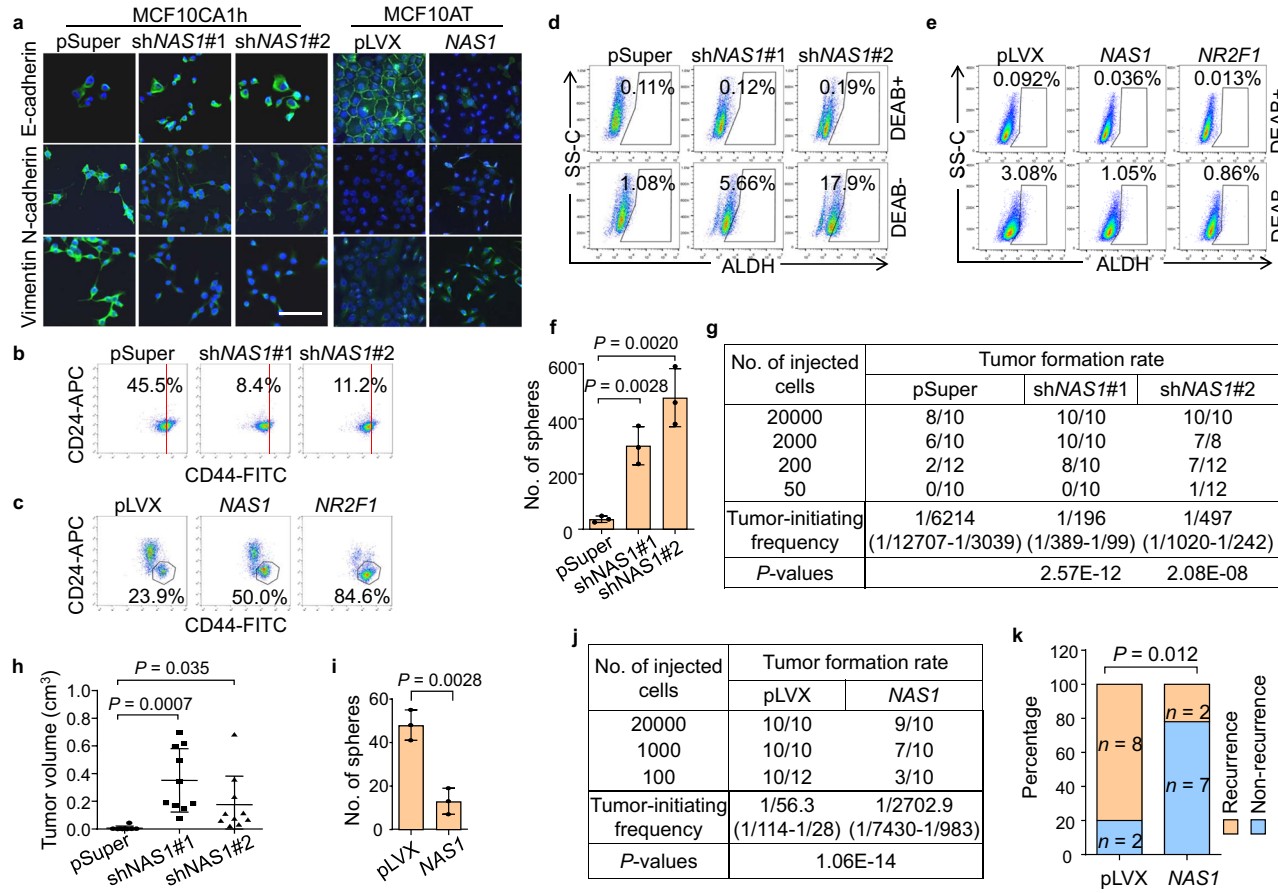

**Fig. 3 NAS1 promotes EMT but inhibits tumorigenicity of breast cancer cells. a** Representative IF images of EMT markers in CA1h-P2 with *NAS1* knockdown (left) or MCF10AT with *NAS1* overexpression (right). Scale bar, 100 µm. Random microscopic fields (*n* > 3) showed similar results. **b**, **c** Flow cytometric analyses of CD24/CD44 expression in CA1h-P2 with *NAS1* knockdown (**b**), or MCF10AT with *NAS1* or *NR2F1* overexpression (**c**). Red vertical lines indicate the same abscissa value in different panels. Numbers denote the CD24−CD44+ percentages. **d**, **e** Flow cytometric analyses of ALDH+ BCSCs in CA1h-P2 with *NAS1* knockdown (**d**) or in MCF10AT with *NAS1* or *NR2F1* overexpression (**e**). Numbers show the ALDH+ percentages. DEAB, the ALDH inhibitor. **f** Tumorsphere formation of CA1h-P2 with *NAS1* knockdown (*n* = 3 culturing experiments). **g** Tumor incidence in NOD/SCID mice orthotopically injected with various numbers of CA1h-P2 cells after *NAS1* knockdown. **h** Tumor sizes of the mouse groups injected with 20,000 cells in (**g**) at week 7 (*n* = 8, 10, and 10 tumors, respectively). **i** Tumorsphere formation of MCF10AT with *NAS1* overexpression (*n* = 3 culturing experiments). **j** Tumor incidence in NOD/SCID mice orthotopically injected with various numbers of *NAS1*-overexpressing MCF10CA1a cells. **k** Tumor recurrence rate 2 weeks after surgically removing tumors at the size of about 1 cm³ in the mice injected with 20,000 cells in (**j**). Data represent mean ± SD (**f**, **h**, **i**). Statistical significance was determined by a two-tailed unpaired *t* test (**f**, **h**, **i**), two-sided chi-square test (**k**, **g**, **j**).

of seeded tumor (Fig. 2f), resulting in persistent single tumor cells in lungs and fewer metastatic tumor nodules in the end (Fig. 2g, h). To further analyze the role of *NAS1* in the entire metastasis process, *NAS1* was overexpressed in 4175-LM2 (Supplementary Fig. 4g), a lung-metastatic subline of MDA-MB-231 breast cancer cells[50], followed by orthotopic injection of the cells into the mammary fat pad of mice. Subsequent analysis showed that *NAS1* enhanced the local invasion of the tumors (Fig. 2i), but resulted in more long-term solitary tumor cell foci and fewer metastatic nodules in the lungs (Fig. 2j, k). These data suggested that *NAS1* promotes the metastatic seeding of tumor cells, but inhibits their proliferation for metastatic outgrowth, leading to the dormancy of breast cancer cells in lungs.

**NAS1 promotes EMT and invasion but inhibits the tumor-initiating capacity of cancer cells.** To analyze why *NAS1* promotes cancer cell seeding but inhibits metastatic outgrowth in the lungs, we firstly analyzed the EMT status and invasiveness of cancer cells. *NAS1* knockdown in CA1h-P2 CSC subpopulation partially induced mesenchymal–epithelial transition (MET), as shown by cellular morphology and expression of various EMT

markers (Fig. 3a and Supplementary Fig. 5c, d), accompanied by decreases in cell migration, invasion, and resistance to anti-proliferative drugs including paclitaxel and doxorubicin (Supplementary Fig. 5e–h). Furthermore, orthotopic xenograft tumors of CA1h-P2 showed less invasion into the para-tumor stroma after *NAS1* knockdown (Supplementary Fig. 5i). In the epithelial MCF10AT cells, *NAS1* overexpression resulted in EMT changes (Fig. 3a and Supplementary Fig. 5j, k), together with elevated migratory and invasive capacities of the cells (Supplementary Fig. 5l, m). Importantly, *NAS1* overexpression in MCF10CA1a followed by orthotopic injection of the cells into mice resulted in more circulating tumor cells in the blood of mice, indicating enhanced tumor dissemination (Supplementary Fig. 5n). In addition, we also observed that *NAS1* knockdown led to CD44 downregulation but ALDH upregulation in CA1h-P2 cells, while *NAS1* overexpression promoted the transition of ALDH+ cells, previously known as the epithelial-like CSCs[28], to the CD44high mesenchymal-like CSCs in MCF10AT (Fig. 3b–e), corroborating the EMT-promoting role of *NAS1*.

Metastatic outgrowth requires the capacity of disseminated cancer cells to initiate a new tumor. Thus, we further assessed the

role of *NAS1* in tumor initiation. *NAS1* knockdown significantly increased the capacity of CA1h-P2 to form three-dimensional tumorspheres (Fig. 3f). More importantly, limiting dilution assays of tumorigenesis showed that *NAS1* knockdown greatly increased the in vivo tumor initiation, as well as tumor growth of CA1h-P2 (Fig. 3g, h). In contrast, *NAS1* overexpression in MCF10AT impaired tumorsphere formation (Fig. 3i). Similar results were also observed in MCF7, in which *NAS1* activation by CRISPR/Cas9 synergistic activation mediator (SAM)[51] reduced tumorsphere formation (Supplementary Fig. 5o). In addition, *NAS1* overexpression in MCF10CA1a (Supplementary Fig. 4g) diminished the tumorigenic capacity of cells in mice (Fig. 3j), as well as the rate of tumor recurrence after surgical removal of the primary tumors (Fig. 3k). Overall, these data suggested that *NAS1* promotes cancer cell traits prerequisite for tumor spreading and dissemination, including EMT, migration, and invasion, but inhibits the capacity to initiate new tumor growth, thus leading to post-dissemination dormancy.

**NAS1 promotes dormancy by upregulating NR2F1.** Then we interrogated the molecular mechanism of *NAS1* in the regulation of metastasis. We firstly focused on the gene *NR2F1*, which is located in proximity to *NAS1* and encodes a transcription factor known to promote tumor dormancy[14,16,52,53]. We found that NR2F1 was also differentially expressed in the MCF10cells with varied metastatic capacities and, importantly, was upregulated by *NAS1* (Fig. 4a). On the contrary, the expression *NAS1* was not affected by *NR2F1* in cancer cells (Supplementary Fig. 6a, b). In addition, *NR2F1* overexpression showed a *NAS1*-mimicking effect to induce cellular EMT (Fig. 4b) and a shift from ALDH$^+$ to CD44$^{high}$ cell population (Fig. 3c, e), together with inhibition of tumorsphere formation (Fig. 4c) in multiple cell lines.

To further validate that NR2F1 acts downstream of *NAS1* in the regulation of metastatic dormancy, *NR2F1* was overexpressed in CA1h-P2 cells with *NAS1* knockdown. NR2F1 efficiently restored the mesenchymal-like phenotype of these cells (Fig. 4d). In addition, *NR2F1* reduced the ALDH$^+$ content and tumorsphere-forming capacity of the cells that were enhanced by *NAS1* knockdown (Fig. 4e, f). More importantly, when CA1h-P2 cells were inoculated into the mice, *NR2F1* overexpression abrogated the effect of *NAS1* knockdown and inhibited tumor cell proliferation after seeding (Fig. 4g), leading to reduced metastatic outgrowth and formation of tumor nodules in lungs (Fig. 4h, i). Reciprocally, *NR2F1* knockdown in MCF10CA1a cells with *NAS1* overexpression resulted in reverse of *NAS1*-induced EMT changes (Supplementary Fig. 7a). Tumorsphere formation and ALDH$^+$ fraction of the cancer cells that were impaired by *NAS1* were also rescued by *NR2F1* knockdown (Supplementary Fig. 7b, c). These data suggested that *NAS1* promotes metastatic dormancy of breast cancer by regulating NR2F1.

**NAS1 and PTBP1 cooperatively enhance the IRES activity of NR2F1-5′UTR.** To further explore how *NAS1* regulates NR2F1 expression, we performed RNA pulldown assays of *NAS1*, followed by mass spectrometry (MS) analyses either directly and after sodium dodecyl sulfate-polyacrylamide gel electrophoresis of the pulldown elution (Supplementary Fig. 8a), to identify the proteins interacting with *NAS1*. Among the *NAS1*-bound proteins (Supplementary Table 2), polypyrimidine tract-binding protein 1 (PTBP1) ranked as a top candidate that was abundantly and consistently enriched by *NAS1* pulldown. *NAS1* pulldown of PTBP1 was verified by Western blotting (WB) (Fig. 5a). Further analyses with various truncations of the *NAS1* sequence showed that it was the 3′ fragment of *NAS1* that was bound to PTBP1 (Supplementary Fig. 8b, c). In addition, *NAS1* RNA was detected

in the immunoprecipitates of PTBP1 by RNA immunoprecipitation (RIP) assays, further demonstrating the binding of *NAS1* to PTBP1 (Fig. 5b). Notably, PTBP1 was expressed in both cytoplasm and nucleus of MCF10CA1h (Supplementary Fig. 8d), a pattern similar to *NAS1* expression (Supplementary Fig. 4d).

Thus we analyzed the function of PTBP1 in NR2F1 expression. Overexpressing *PTBP1* in MCF10AT upregulated NR2F1 while interfering *PTBP1* by siRNAs obviously decreased the protein level of NR2F1 in MCF10CA1h (Supplementary Fig. 8e), implying the involvement of PTBP1 in *NAS1* regulation of NR2F1 expression. Indeed, *PTBP1* knockdown in MCF10AT with *NAS1* overexpression completely blocked the upregulation of NR2F1 by *NAS1*, while *PTBP1* overexpression in CA1h-P2 with *NAS1* knockdown recovered the protein level of NR2F1 (Fig. 5c), affirming that PTBP1 mediates the function of *NAS1* in NR2F1 regulation.

Then we analyzed how PTBP1 regulates NR2F1 expression. Previous studies showed that PTBP1 could bind to gene promoter regions to regulate transcription[54], or bind to the polypyrimidine regions of mRNAs to regulate post-transcriptional processes including alternative splicing[55,56], mRNA decay[57], and IRES-dependent translation[58–60]. We first tested the effect of PTBP1 and *NAS1* on NR2F1 promoter via luciferase reporter assays and found that NR2F1 promoter activity was not obviously changed by either PTBP1 or *NAS1* (Supplementary Fig. 8f). Chromatin immunoprecipitation (ChIP)-qPCR analysis also detected no binding of PTBP1 on NR2F1 promoter (Supplementary Fig. 8g), hence excluding the possibility of PTBP1 regulation of NR2F1 promoter activity.

Alternatively, RIP assays showed that NR2F1 mRNA was bound by PTBP1 (Fig. 5d). The MS2–GFP fusion protein immunoprecipitating assays of MS2-tagged *NAS1* also showed the binding of *NAS1* RNA to NR2F1 mRNA (Fig. 5e). Truncation analysis of *NAS1* further revealed the preferential binding of NR2F1 mRNA to the 5′ fragment of *NAS1* RNA (Supplementary Fig. 8b, h). Together with the above observations of PTBP1–*NAS1* interaction (Fig. 5a, b), these data demonstrated the interactions between each pair of PTBP1 protein, *NAS1* RNA, and NR2F1 mRNA. In addition, we found that *NAS1* knockdown impaired the binding of PTBP1 and NR2F1 mRNA (Fig. 5f and Supplementary Fig. 8i). In contrast, *PTBP1* inhibition led to no obvious changes in the amount of NR2F1 mRNA pulled down by *NAS1* (Fig. 5g and Supplementary Fig. 8j). Thus, these data indicated that *NAS1* promotes the binding of PTBP1 and NR2F1 mRNA.

We then analyzed how *NAS1* and PTBP1 regulate NR2F1 mRNA. We identified four possible splicing variants of NR2F1 in the RNA-sequencing data of CA1h-P2 cells, but the expression levels of none of these variants were obviously changed by *NAS1* knockdown (Supplementary Fig. 8k), excluding the possibility that *NAS1* regulates alternative splicing of NR2F1 mRNA. We also tested the stability of NR2F1 mRNA by treating cells with actinomycin D and found no effects by *NAS1* overexpression or knockdown (Supplementary Fig. 8l). Then we noticed multiple polypyrimidine regions in the 5′UTR of NR2F1 mRNA and thus speculated that PTBP1 and *NAS1* might facilitate the IRES function of NR2F1-5′UTR. To verify this hypothesis, we constructed an IRES activity reporter plasmid (Fig. 5h) and cloned the full 1.8 kb 5′UTR sequence upstream of AUG of NR2F1 mRNA and its two truncations, the 0.8 kb polypyrimidine region (termed as 5′UTR-PT) and the rest 1.0 kb region that is GC-rich (termed as 5′UTR-GC) (Supplementary Table 3) into the plasmid (Fig. 5h). IRES reporter assays revealed that the full 5′UTR had a mild activity to initiate translation. Interestingly, 5′UTR-PT showed a much stronger IRES activity. On the contrary, 5′UTR-GC has no IRES activity at all (Fig. 5i). To confirm that

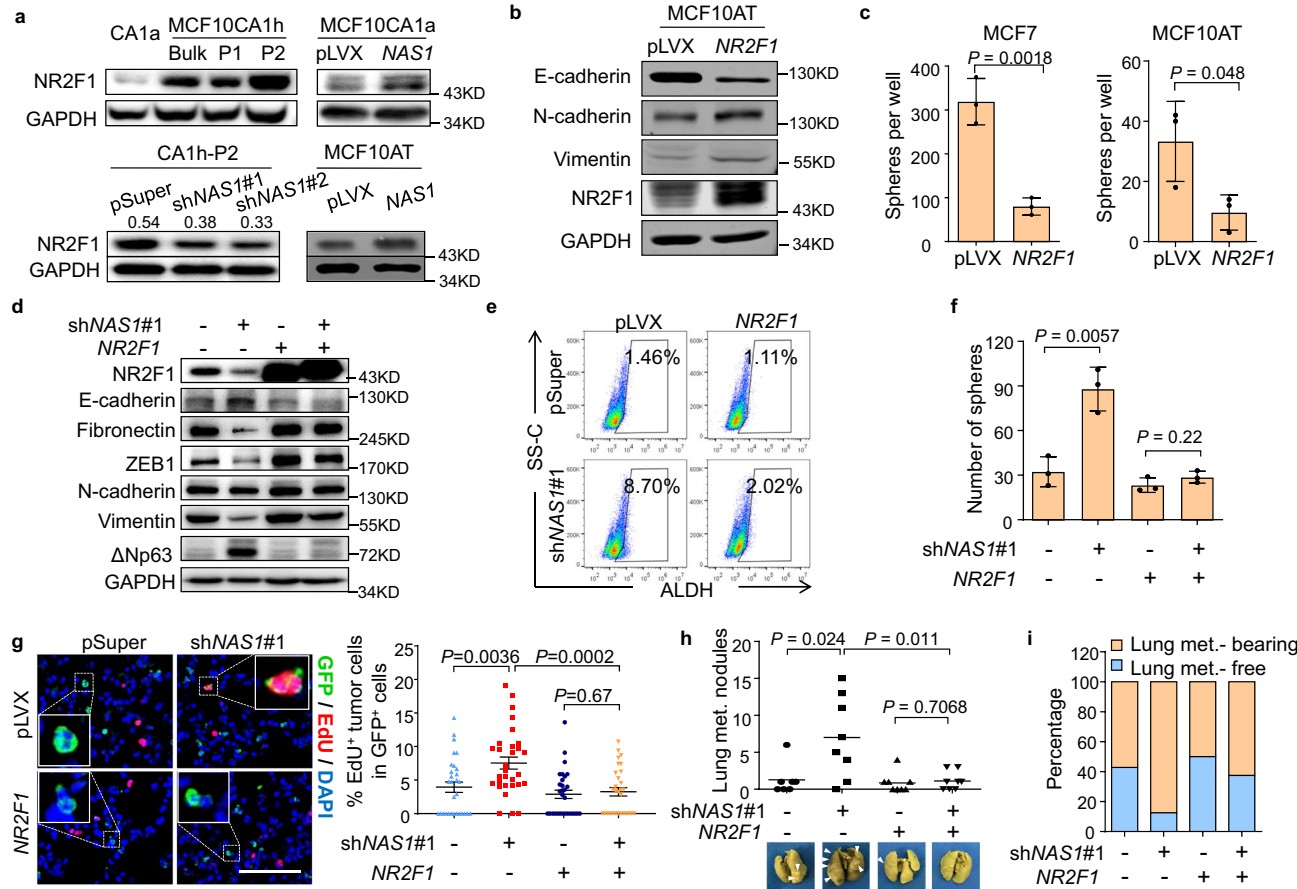

**Fig. 4 NAS1 suppresses breast cancer lung metastasis by NR2F1. a** NR2F1 expression in MCF10CA1a, MCF10CA1h, P1, and P2, or after *NAS1* knockdown or overexpression. Relative quantitation of NR2F1 expression after *NAS1* knockdown was shown for CA1h-P2 cells. **b** Expression of EMT markers after *NR2F1* overexpression in MCF10AT. **c** Tumorsphere formation of MCF7 and MCF10AT after *NR2F1* overexpression (n = 3 culturing experiments). **d–i** Analyses of CA1h-P2 cells with *NAS1* knockdown and/or *NR2F1* overexpression. Shown are protein levels of EMT markers (**d**), flow cytometric analyses of the ALDH⁺ BCSCs (**e**), tumorsphere formation (**f**, n = 3 culturing experiments), representative IF images (left), and quantitation of EdU⁺ tumor cells (right) in lungs 4 days after intravenous injection of the cells (**g**, n = 30 RMFs from 3 mice), lung surface nodules 18 weeks after intravenous injection (**h**, n = 7, 8, 8 and 8 mice, respectively), and the lung metastasis incidence (**i**). The triangles indicate metastatic nodules on the lung surface (**h**). Scale bar, 100 μm. Data represent mean ± SD (**c**, **f**), mean ± SEM (**g**) or mean with single points (**h**). Arrows point to tumor nodules in (**h**). Statistical significance was determined by a two-tailed unpaired *t* test. Experiments in **a**, **b**, **d**, **e** were repeated at least three times independently with similar results; data from one representative experiment are shown.

such an effect of 5′UTR-PT to activate the downstream gene is due to regulation of translation rather than transcription, we also assessed the potential promoter activity of these regions using the pGL3 reporter. Although mild promoter activities of 5′UTR and 5′UTR-GC were detected, 5′UTR-PT had no ability to initiate transcription (Supplementary Fig. 8m). Therefore, these data suggested that the polypyrimidine region of *NR2R1* 5′UTR possesses an innate IRES activity, while the existence of downstream GC-rich sequence hinders such activity for mRNA translation.

Interestingly, bioinformatic analyses with protein–RNA and RNA–RNA interaction-predicting tools (catRAPID[61] and IntaRNA[62–65]) suggested that PTBP1 and *NAS1* bind to the polypyrimidine and GC-rich regions, respectively, of *NR2F1*-5′ UTR (Fig. 5j). RIP and RNA pulldown assays with various *NR2F1* mRNA areas confirmed that *NAS1* RNA preferentially bound to 5′UTR-GC (Fig. 5k), and PTBP1 specifically interacted with 5′ UTR-PT but not other regions, including 5′UTR-GC, CDS or 3′ UTR of *NR2F1* mRNA (Fig. 5l). Importantly, *PTBP1* inhibition led to the suppression of 5′UTR-PT IRES activity (Fig. 5m). Reciprocally, *NAS1* overexpression could enhance IRES activity of 5′UTR, but such regulation is dependent on the presence of the

GC-rich region. The strong IRES activity of 5′UTR-PT was not affected by *NAS1* (Fig. 5n). Taken together, these data indicated that PTBP1 binds to the polypyrimidine region of *NR2F1*-5′UTR to initiate IRES activity, but the *NR2F1* translation process is suppressed by the downstream GC-rich region. Such suppression can be relieved by the binding of *NAS1*, probably by remodeling of the RNA structure formed in this GC-rich area (Fig. 5o).

**NR2F1 represses the expression of ΔNp63**. We continued to analyze the downstream molecular events of *NAS1*–NR2F1 to promote tumor cell dormancy, by RNA-sequencing analysis of cancer cells after *NAS1* knockdown or *NR2F1* overexpression. Among the genes commonly regulated by both *NAS1* and *NR2F1* (Fig. 6a and Supplementary Table 4), the microRNA miR-205 and the *TP63* gene variant *ΔNp63*, which is a transcriptional regulator of miR-205[66,67], were noticed. miR-205 is known to suppress EMT of tumor cells by targeting ZEB1 and SIP1[68]. *ΔNp63* is a marker of epithelial stem cells with well-studied functions to regulate cell stemness by activating signaling pathways including WNT[69] and NOTCH[70] and to maintain the epithelial feature of cells by activating miR-205[67]. Our analyses showed that expression of both miR-205 and *ΔNp63* was

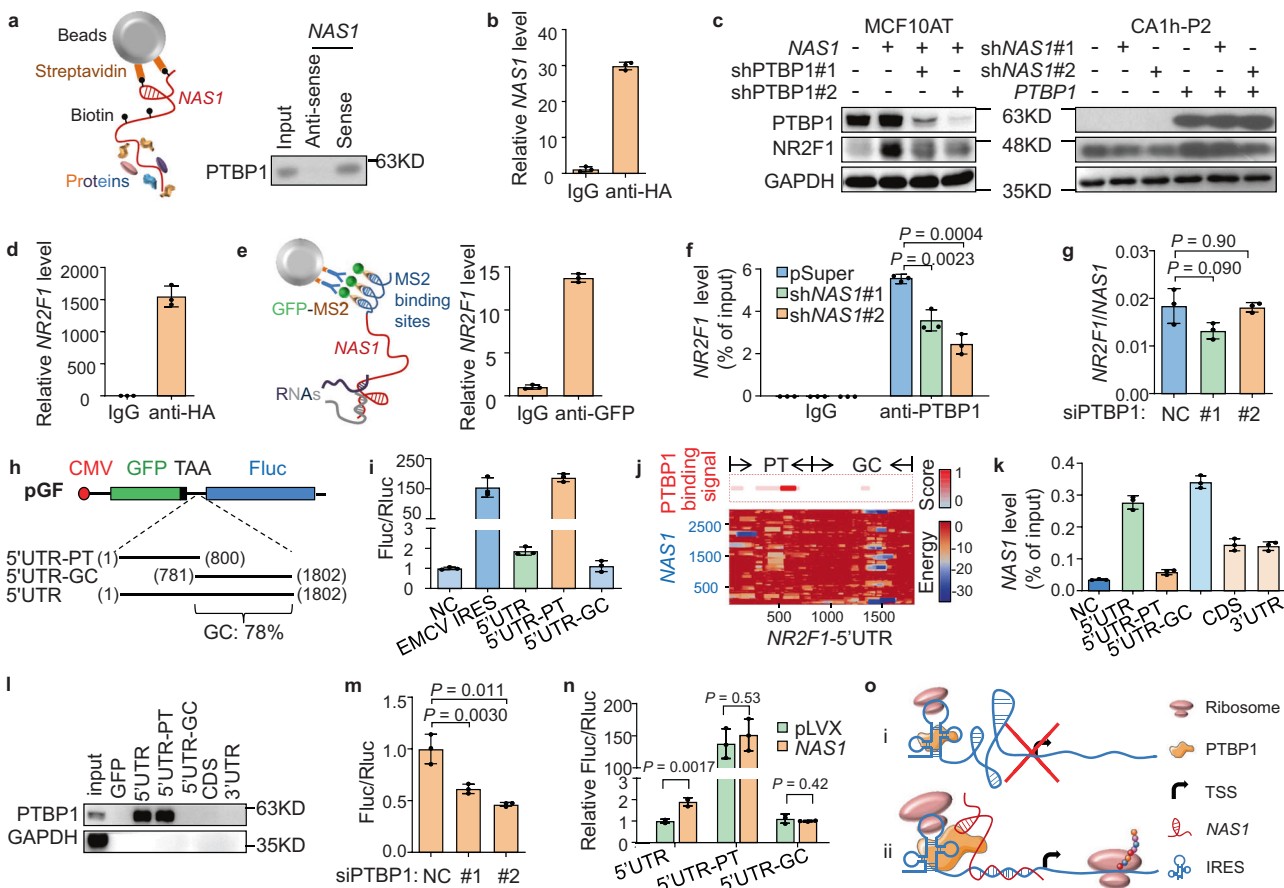

**Fig. 5 NAS1 and PTBP1 cooperatively enhance the IRES function of NR2F1-5′UTR. a** PTBP1 pulldown by *NAS1* RNA in MCF10CA1h lysates. A schematic of the RNA pulldown assay was shown on the left. **b** *NAS1* RNA level in RIP assays for PTBP1-interacting RNAs in MCF10CA1h expressing PTBP1-HA. **c** NR2F1 protein level after *PTBP1* knockdown in *NAS1*-overexpressing MCF10AT (left) or after *PTBP1* overexpression in CA1h-P2 with *NAS1* knockdown (right). **d** NR2F1 mRNA level in RIP assay for PTBP1-interacting RNAs in MCF10CA1h expressing PTBP1-HA. **e** RIP assays with the MS2–GFP system to analyze the RNA–RNA interaction between *NAS1* and *NR2F1*. A schematic of the assay was shown on the left. **f** RIP analyses of the binding of PTBP1 to *NR2F1* mRNA in CA1h-P2 with *NAS1* knockdown ($n = 3$ replicates from one experiment). **g** RIP analyses of RNA–RNA interaction between *NAS1* and *NR2F1* in MCF10CA1h with *PTBP1* knockdown. The ratios of *NR2F1* to *NAS1* were shown ($n = 3$ replicates from one experiment). **h** Schematic of the pGF plasmid and the *NR2F1*-5′UTR segments for IRES activity analyses. Numbers in parentheses indicate the start and end base-pair positions of the segments. GFP and firefly luciferase are transcribed by the same CMV promoter, while luciferase translation is dependent on the IRES activity of the insert sequence between the two genes. **i** IRES activity of *NR2F1*-5′ UTR was analyzed by a dual-luciferase reporter assay of the pGF system ($n = 3$ wells from one experiment). EMCV IRES was used as a positive control. NC, negative control. **j** PTBP1 binding sites on *NR2F1*-5′UTR predicted by catRAPID (top), and *NAS1* binding sites on *NR2F1*-5′UTR predicted by IntaRNA (bottom). **k** RIP analyses of the binding of *NAS1* to different regions of *NR2F1* mRNA. **l** RNA pulldown assays for interaction between PTBP1 and regions of *NR2F1* mRNA. **m** Effect of PTBP1 on IRES activity of *NR2F1*-5′UTR-PT ($n = 3$ wells from one experiment). **n** Effect of *NAS1* overexpression on IRES activity of *NR2F1*-5′UTR ($n = 3$ wells from one experiment). **o** Schematic of the mechanism by which PTBP1 and *NAS1* regulate the IRES activity of *NR2F1*-5′UTR. PTBP1-mediated *NR2F1* translation was halted by the GC-rich segment (i) and could be activated by the binding of *NAS1* (ii). TSS transcription start site. Data represent mean ± SD. Statistical significance was determined by a two-tailed unpaired *t* test. Experiments were repeated at least three times independently with similar results; data from one representative experiment are shown.

significantly suppressed by *NAS1* and *NR2F1* (Fig. 6a), which was further confirmed by qPCR and WB assays (Fig. 6b–d). In addition, NR2F1 completely blocked the upregulation of ΔNp63 by *NAS1* knockdown in CA1h-P2 cells (Fig. 4d). PTBP1 also suppressed the expression of ΔNp63 (Supplementary Fig. 8e).

Thus, we investigated the role of ΔNp63 in *NAS1*-induced EMT and suppression of tumorigenicity. ΔNp63 inhibition by a reported shRNA[69] in CA1h-P2 cells with *NAS1* knockdown reversed the MET phenotype, leading to regain of mesenchymal morphology and marker expression (Fig. 6e and Supplementary Fig. 9a). Tumorsphere formation that was enhanced by *NAS1* knockdown was also effectively suppressed by ΔNp63 shRNA (Fig. 6f). Furthermore, GSEA analysis of control and *NR2F1*-overexpressing MCF7 cells showed that the previously identified gene sets downregulated and upregulated by ΔNp63[71]

(Supplementary Table 4) were enriched in the cells with or without *NR2F1* overexpression, respectively (Fig. 6g), indicating that NR2F1 adversely affects the downstream genes of ΔNp63. ChIP-qPCR analysis showed the binding of NR2F1 on ΔNp63 promoter (Fig. 6h). NR2F1 inhibition led to the activation of ΔNp63 promoter activity (Supplementary Fig. 9b), suggesting the direct regulation of ΔNp63 by NR2F1. Altogether, these data demonstrated that NR2F1 mediates the dormant-promoting function of *NAS1* by inhibiting the expression of ΔNp63.

In addition, we found that ΔNp63 activated miR-205 expression while ΔNp63 knockdown inhibited miR-205 expression in breast cancer cells (Supplementary Fig. 9c, d), corroborating previous reports showing that ΔNp63 transcriptionally regulates miR-205[66]. Importantly, *NAS1* and *NR2F1* inhibited the expression of miR-205, while the effect of *NAS1* on miR-205 was effectively blocked

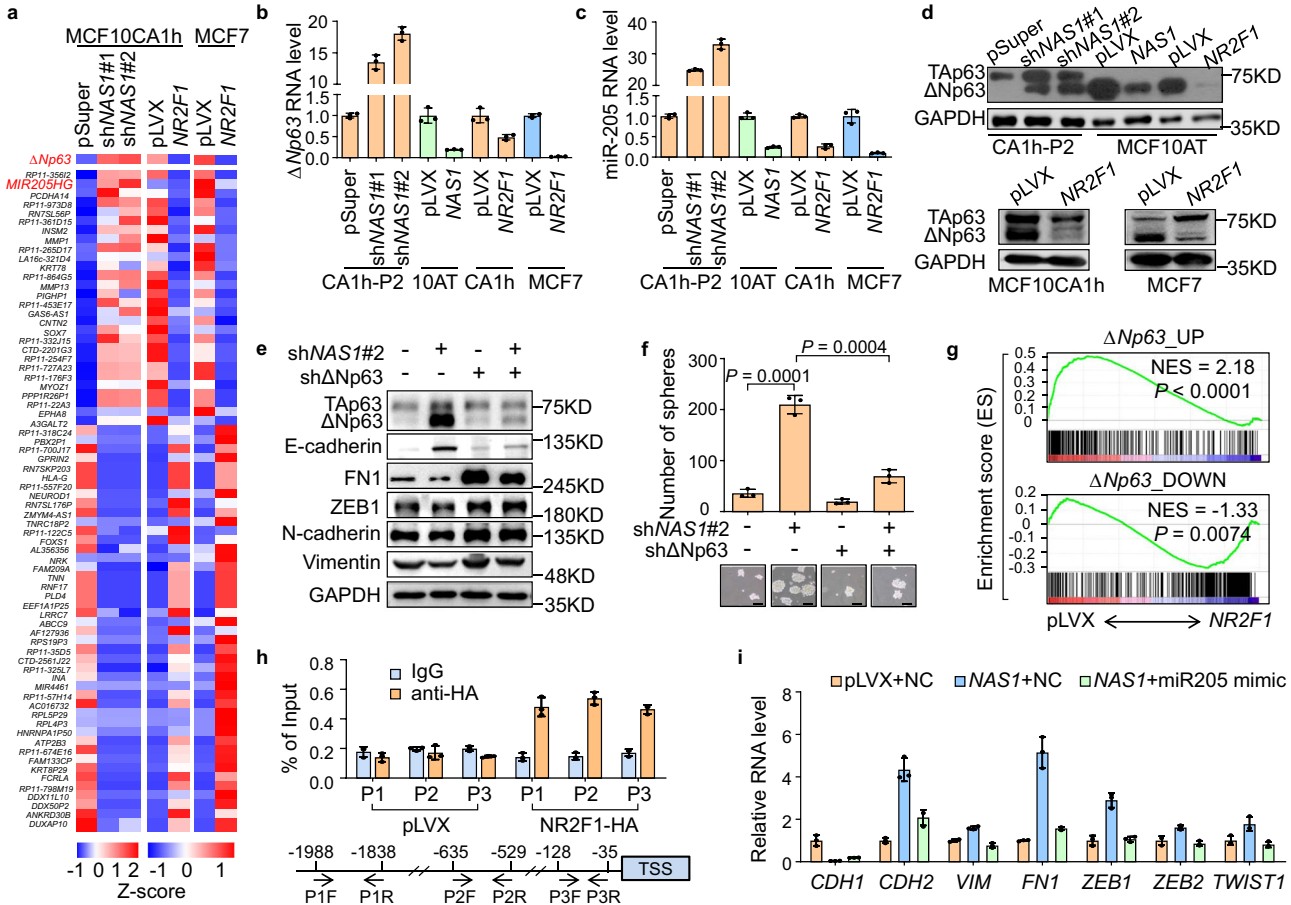

**Fig. 6 NR2F1 mediates the role of *NAS1* in dormancy by inhibiting *ΔNp63* transcription. a** Expression heatmap of genes affected by *NAS1* and NR2F1 in both MCF10CA1h and MCF7. **b** qPCR verification of *ΔNp63* regulation by *NAS1* and NR2F1. **c** qPCR verification of miR-205 regulation by *NAS1* and NR2F1. **d** ΔNp63 protein levels regulated by *NAS1* or NR2F1. **e, f** EMT marker protein expression (**e**) and tumorsphere formation (**f**, $n = 3$ culturing experiments) after *ΔNp63* knockdown in CA1h-P2 cells with *NAS1* knockdown. **g** GSEA analyses of the gene set upregulated and downregulated by *ΔNp63* in MCF7 cells with *NR2F1* overexpression versus control. **h** ChIP-qPCR analysis of NR2F1 binding to *ΔNp63* promoter ($n = 3$ replicates from one experiment). P1–3, primer pairs #1–3. **i** mRNA levels of EMT markers in MCF10AT with *NAS1* overexpression and/or miR-205 mimic. Data represent mean ± SD. Statistical significance was determined by a two-tailed unpaired *t* test. Experiments were repeated at least three times independently with similar results; data from one representative experiment are shown. Scale bar, 100 μm.

by *ΔNp63* (Supplementary Fig. 9c–e), indicating that *NAS1* regulates miR-205 by suppressing *ΔNp63*. Furthermore, miR-205 inhibition restored the mesenchymal phenotype of CA1h-P2 cells with *NAS1* knockdown (Supplementary Fig. 9f), while treating *NAS1*-overexpressing MCF10AT with miR-205 mimics induced epithelial restoration of the cells (Fig. 6i and Supplementary Fig. 9g). In addition, miR-205 inhibition also blocked the MET changes of CA1h-P2 cells that were induced by *NR2F1* knockdown (Supplementary Fig. 9h). Therefore, miR-205 acts downstream of *NAS1*–NR2F1–ΔNp63 signaling axis to regulate cellular EMT.

**NAS1 correlates with NR2F1, EMT markers, and reduced metastasis in human breast tumors.** Finally, we assessed the clinical significance of *NAS1* expression in clinical breast tumor samples. First, a positive correlation of *NAS1* RNA and NR2F1 protein levels was observed in tumor samples of breast cancer patients (Fig. 7a), supporting the role of *NAS1* in regulating NR2F1. Then, by analyzing the RNA sequencing data of the TCGA breast cancer cohort, we found that *NAS1* significantly correlates with a wide array of EMT-related genes (Fig. 7b). A positive correlation of *NAS1* with *NR2F1* expression and *ΔNp63* with miR-205 was also observed in the TCGA pan-cancer cohort,

while the expression of *NR2F1* was negatively correlated with the expression of both *ΔNp63* and miR-205 (Supplementary Table 5). In the Fudan triple-negative breast cancer cohort[47], *NAS1* was also positively correlated with enrichment of the previously reported M-BCSC[28] and EMT[72] gene signatures, as well as a deficit of the E-BCSC signature[28], as shown by single-sample GSEA (ssGSEA)[73] (Fig. 7c). Next, analysis of 89 breast cancer samples collected from Qilu Hospital revealed that high *NAS1* expression of the tumors was associated to a lower risk of metastasis and tumor relapse (Fig. 7d, e). Meanwhile, analyses of the Kaplan–Meier Plotter clinical database[74] also displayed the link of *NAS1* to improved recurrence-free survival of breast cancer patients (Supplementary Fig. 10a–d). Concordantly, the expression of the gene sets upregulated by *NAS1* knockdown or *NR2F1* overexpression, identified by our transcriptomic sequencing analyses (Supplementary Table 4), was linked to accelerated and decelerated metastasis, respectively, in the Netherlands Cancer Institute breast cancer cohort[75] (Fig. 7f, g). Immunostaining of a breast cancer tissue microarray also revealed that NR2F1 expression was linked to improved relapse-free survival of the patients (Supplementary Fig. 10e, f). Finally, we also observed downregulation of *NAS1* in breast tumors compared with the normal mammary tissues in the TCGA cohort (Supplementary

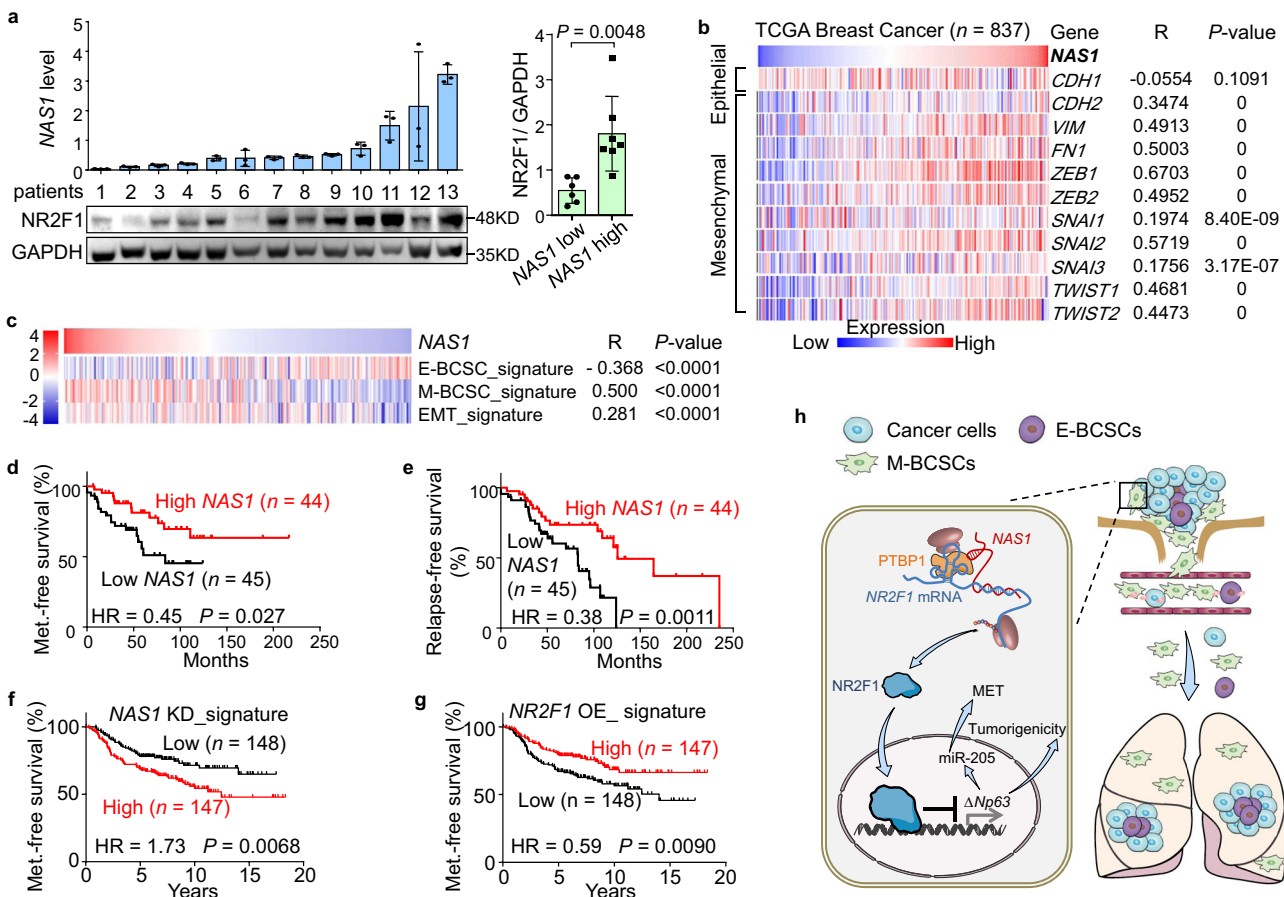

**Fig. 7 The clinical relevance of *NAS1* expression in human breast tumors. a** Expression of *NAS1* (upper left) and NR2F1 (lower left) in breast tumors from 13 patients, and quantitation of NR2F1 protein levels in tumors with high or low *NAS1* expression (right) (*n* = 13 patients). **b** Correlation analyses between *NAS1* and EMT-related genes in 837 TCGA breast cancer samples. R, Pearson's correlation coefficient of the genes to *NAS1*. **c** Correlation between *NAS1* expression and the ssGSEA scores of E-BCSC, M-BCSC, and EMT signatures in the Fudan cohort[47] (*n* = 447 patients). **d** Metastasis-free survival analysis of the Qilu breast cancer cohort by *NAS1* expression. HR hazard ratio. **e** Relapse-free survival analysis of the Qilu cohort by *NAS1* expression. **f**, **g** Metastasis-free survival analyses of NKI cohort[75] (*n* = 295 patients) by ssGSEA scores of gene sets regulated by *NAS1* knockdown (**f**) or *NR2F1* overexpression (**g**). **h** Schematic model of the mechanism by which *NAS1* induces metastatic dormancy of breast cancer cells via *NAS1*/NR2F1/ΔNp63 axis. Data represent mean ± SD (**a**). Statistical significance was determined by a two-tailed unpaired *t* test (**a**), paired *t* test (**b**, **c**), or two-sided log-rank test (**d**–**g**).

Fig. 10g), supporting a role of *NAS1* to suppress tumorigenicity. In summary, these data suggested that *NAS1* correlates with expression of NR2F1 and EMT markers, and links to better prognosis in human clinical breast cancer.

## Discussion

Previously *NAS1* has been reported by several studies to regulate proliferation and migration of different cancer types, including hepatocellular carcinoma[40], endometrial cancer[41], thyroid cancer[42,43], and osteosarcoma[44]. These functions of *NAS1* have been attributed to its microRNA-sponging function. A most recent study[39] also demonstrated the link of *NAS1* expression to late recurrence of ER+ breast cancer. Here we not only report a role of *NAS1* to promote metastatic dormancy of breast cancer, consistent with the recent study[39] but also reveal a previously unidentified functional mechanism of *NAS1*, by which *NAS1* binds to the GC-rich 5'UTR area of *NR2F1* mRNA and recruits PTBP1 to enhance the IRES-dependent translation of NR2F1. Further NR2F1 suppresses *ΔNp63* expression, leading to cellular EMT changes and impaired tumorigenicity (Fig. 7h). NR2F1 plays a well-established role in dormancy. It has been shown to induce cancer cell quiescence and tumor dormancy in various cancer types including breast cancer, prostate cancer, and head

and neck squamous cell carcinoma[14,16,53]. Clinical analyses also demonstrated the correlation of *NR2F1* expression to the dormancy of human tumors[14,52]. Our study corroborates these reports and reveals a unique mechanism of NR2F1 in metastatic dormancy. Notably, our data indicate that both *NAS1* and NR2F1 play dual roles in cancer to promote tumor invasion but inhibit proliferation, which is seemingly conflicting for cancer progression but actually both contribute to the accumulation of dormant DTCs in metastatic organs. The link of *NAS1* expression to the improved prognosis of cancer patients is likely attributed to its anti-proliferation role, which results in tumor suppression in the primary site and metastatic dormancy in secondary organs, but not the pro-dissemination effect of *NAS1*. These findings will deepen our understanding of the roles of lncRNAs and NR2F1 in metastatic dormancy.

Importantly, our study demonstrates a scenario, by mesenchymal-like tumor cells in particular, of dormancy amid the multi-step process of metastasis. These cells are equipped with all the traits for leaving the primary tumors and arriving at the new organs, including migration, invasion, and resistance to adjuvant chemotherapy, but they are less capable to initiate a new tumor and colonize the target organs. This will lead to the accumulation of solitary tumor cells or small foci in secondary organs, i.e., metastatic dormancy. Other types of cells, either not

capable of tumor dissemination (such as MCF10AT) or prone to active proliferation once reaching the new sites (such as MCF10CA1a), will not demonstrate such a dormant stage of metastasis. It has been speculated that EMT may cause dormancy[30]. Our data provide the evidence for such speculation and also recall previous studies showing that metastatic colonization of mesenchymal tumor cells requires the reversal to epithelial type[24,25,76] and that epithelial-like circulating tumor cells are the culprit of tumor metastasis[77,78].

Our study will also help delineate the role of CSCs in metastatic dormancy. As metastatic growth from single DTCs is essentially the tumor-initiating feature of CSCs, activation from dormancy is often thought to be accompanied by gain of stem-like characteristics[8,12]. However, some studies also indicate dormant cells are CSCs due to the similar phenotypes and shared mechanisms of these two cell types in slow cell cycle, drug resistance, and long-term survival[22,23]. This apparent contradiction could be reconciled by the heterogeneity of CSCs. Our data suggest that the mesenchymal-like $CD24^-CD44^{high}$ BCSCs are prone to tumor dissemination and dormancy, while the epithelial-like $ALDH^+$ BCSCs are metastasis-competent and represent awakening from dormancy. This model is also consistent with previous findings that dormant breast cancer cells are enriched of the $CD44^{high}CD24^{low}$ population[20] and that the $CD44^{high}CD24^{low}$ profile defines the mesenchymal type of BCSCs which are relatively less tumorigenic than the $ALDH^+$ BCSCs[27,28]. In addition, tumor initiation and drug resistance have been often considered as the two defining features of CSCs. However, studies of CSC heterogeneity indicate that these two features might be separate in different CSC subsets, although it could still hold true that CSCs as a whole population is more aggressive in both tumorigenicity and drug resilience than non-CSCs. This is particularly noteworthy to reconsider how to define CSCs using in vitro or in vivo assays, and how to target CSCs for precision medicine.

The deadly metastasis results from the accumulation of multiple aggressive features of tumor cells including proliferation and motility. However, previous seminal studies[79–81] have shown that these two features may not be simultaneously present, or even be mutually exclusive in the same tumor cells. Our data re-emphasize such a notion and unravel a dual-role molecular route for such mutual exclusivity. Importantly, the segregation of proliferative and migratory features of tumor cells might underlie the phenomenon of metastatic dormancy in which migratory cancer cells cease metastatic growth after dissemination, while reactivation from dormancy requires the switch of tumor cells back to the proliferative state. Understanding the dynamics of proliferative and migratory states of tumors may provide opportunities to target dormant cancer cells and keep them from regaining the growing momentum.

## Methods

**Constructs and reagents.** The sequences of primers used in this work were provided in Supplementary Table 6. The full length of human *NR2F1-AS1* (*NAS1*) was determined by RACE and cloned into the pLVX-CMV-PGK-puro vector, and the open reading frames of *NR2F1*, *PTBP1* and *ΔNp63* with the HA tag at 3′ end were inserted into pLVX-CMV-IRES-puro/blasticidin for overexpression. For *NAS1* knockdown, shRNAs were designed with the online software *sfold*[82], then the annealed sense and antisense oligonucleotides were cloned into the BglII and HindIII sites of pSuper-Retro-hygro (Oligo Engine). For *PTBP1* and *ΔNp63* knockdown, shRNAs were obtained from the MISSION shRNA library (Sigma) and a previous report[69], respectively, then annealed and cloned into the pLKO.1.puro/blasticidin vectors. *PTBP1*-siRNAs, miR-205 mimics and inhibitors were purchased from GenePharma. The sequences of all siRNAs and shRNAs used in this study are provided in Supplementary Table 6. The pGF plasmid was modified from pLVX-CMV-IRES-puro vector by cloning *GFP* into the EcoRI and XbaI sites and firefly luciferase (*Fluc*) into the BamHI and MluI sites. The *NR2F1*-5′UTR (1,802 bp) sequence was inferred according to the NCBI database and its two segments 5′UTR-PT (1–800 bp) and 5′UTR-GC (781–1802 bp) were inserted

between *GFP* and *Fluc* of pGF vector for IRES activity assays. For promoter activity analysis, the −2000 to +691 sequence flanking the transcription start site of *NR2F1*, as well as *NR2F1*-5′UTR, 5′UTR-PT, and 5′UTR-GC segments, was cloned into pGL3-basic (Promega) with MluI and BglII sites. All cell lines were tested as *Mycoplasma* free.

For WB, flow cytometry, RIP, ChIP, immunohistochemistry (IHC), and IF analyses, the following antibodies were used: APC mouse anti-human CD24 (Biolegend, 311118, 5 μl/$10^6$ cells/100 μl staining volume), FITC mouse anti-human CD44 (BD Pharmingen, 555478, 20 μl/$10^6$ cells/100 μl staining volume), PE mouse anti-human CD44 (BD Pharmingen, 555479, 20 μl/$10^6$ cells/100 μl staining volume), chicken polyclonal GFP antibody (Abcam, ab13970, ICC: 1/1000), Anti-GFP antibody (Abcam, ab290, RIP: 1 μl/$10^6$ cells), donkey anti-Chicken IgY (FITC) secondary antibody (Invitrogen, SA172000, 1/500), E-Cadherin (24E10) rabbit mAb (Cell Signaling technology, 3195S, IF: 1/200, WB: 1/2000), N-Cadherin mouse antibody (BD Pharmingen, 610920, IF: 1/100, WB: 1/1000), Fibronectin mouse antibody (Santa Cruz, sc-59826, WB: 1:1000), Vimentin (D21H3) XP® rabbit mAb (Cell Signaling technology, 5741S, IF: 1/100, WB: 1/1000), ZEB1 rabbit polyclonal antibody (proteintech, 21544-1-AP, WB: 1/1000), mouse anti-Twist antibody (Abcam, ab50887, WB:1/2000), Alexa Fluor 488 donkey anti-rabbit IgG (Invitrogen, A-21206, 1/1000), Alexa Fluor 555 donkey anti-mouse IgG (Invitrogen, A31570, 1/500), rabbit anti-human GAPDH (Merck/Millipore, SAB2103104, WB: 1/5000), rabbit anti-COUP TF1 (NR2F1) antibody (Abcam, ab181137, IHC: 1/200, WB: 1/1000), p63-α (D2K8X) XP® rabbit mAb (Cell Signaling technology, 13109S, WB: 1/1000), PTBP1 (E4I3Q) rabbit mAb (Cell Signaling technology, 57246, WB: 1/1000), Lamin A/C rabbit polyclonal antibody (proteintech, cat#10298-1-AP, WB: 1/1000), rabbit anti-HA (Cell Signaling technology, 3724S, RIP and ChIP: 1/50), HRP-conjugated Goat Anti-mouse IgG (Merck/Millipore, 401,215, 1/10,000), HRP-conjugated Goat Anti-rabbit IgG (Merck /Millipore, 401315, 1/10,000), normal rabbit IgG (Cell Signaling technology, 2729S, RIP and ChIP: 1/50).

**Real-time quantitative PCR (qPCR).** Total RNA was extracted by Trizol reagent (Invitrogen, 15596018) and reverse transcribed by Primescript Reverse transcriptase (Takara, D2680C). Then, FastStart Universal SYBR Green Master (Roche, 4913914001) was used for qPCR detection. Briefly, poly(A) tailing was performed with *Escherichia coli* poly(A) polymerase (NEB, M0276S) before reverse transcription with an RT primer. Then qPCR was performed with primers designed with miRprimer[83]. All primers used in this study, including those for miRNA reverse transcription and qPCR, were provided in Supplementary Table 6.

**Northern Blot.** Total RNA was extracted from CA1h-P2 cells using the Trizol reagent (Invitrogen, 15596018). The *NAS1* probe was obtained by in vitro transcription, and the transcription template was amplified with primers probe-F and probe-R (see primer sequences in Supplementary Table 6), and the obtained fragment (approximately 900 bp) was cloned into the vector pcDNA3.1. Then template fragment containing the T7 promoter was amplified by polymerase chain reaction (PCR) with primers T7-F and probe-R. A 20 μg RNA was loaded for blotting with the DIG Northern Starter Kit (Roche, 12039672910).

**RNA immunoprecipitation.** RIP assays by PTBP1 were performed in MCF10CA1h with PTBP1-HA overexpression or CA1h-P2 with *NAS1* knockdown using HA antibody or PTBP1 antibody, respectively. For RNA–RNA interaction, the MS2 binding site-tagged RNA was expressed in MCF10CA1h with stable MS2–GFP overexpression, and the GFP antibody was used for RIP. Cells were collected and resuspended in polysome lysis buffer (100 mM KCl, 5 mM MgCl₂, 10 mM HEPES, 0.5% Nonidet P-40, 100 U/mL Recombinant RNase Inhibitor, 400 μM ribonucleoside vanadyl complex, 1 mM DTT), and were lysed for 15 min at 4 °C, then the supernatant was collected after centrifuging. Next, the supernatant was divided into two equal parts, and control IgG and antibodies were added for overnight incubation at 4 °C. At the same time, protein A beads (GE Healthcare Life Sciences, 17046901) were washed by NT2 buffer (50 mM Tris-HCl pH 7.4, 150 mM NaCl, 1 mM MgCl₂, 0.05% Nonidet P-40, 0.1 mg/mL bovine serum albumin (BSA)) and were added to each sample following continued incubation for 3 h. After washing beads 4 times with NT2 buffer, 500 μL Trizol reagent was directly added to the beads for RNA extraction. Finally, RNA was reverse transcribed for qPCR detection.

**Chromatin immunoprecipitation (ChIP).** The ChIP assays were performed for the *NR2F1* and *ΔNp63* promoters bound by PTBP1 and NR2F1, respectively. Briefly, HeLa cells were transfected with PTBP1-HA for ChIP of *NR2F1* promoter, and MCF10CA1h cells stably expressing NR2F1-HA were used for ChIP of *ΔNp63* promoter. Briefly, >$10^7$ cells were crosslinked with 1% formaldehyde and quenched by 125 mM glycine. The cell membrane was destroyed and the nucleus was collected. Cell nuclear lysate was sonicated and incubated with control IgG or anti-HA antibody for immunoprecipitation. The complex was captured and precipitated by protein A agarose beads. Captured genomic DNA was reverse-crosslinked and purified by ethanol precipitation with Dr. GenTLE Precipitation Carrier (Takara, 9094). Purified genomic DNA was used for qPCR analysis. The qPCR primers used in these assays are provided in Supplementary Table 6.

**RNA pull-down**. *NAS1* and segments of *NR2F1* mRNA were transcribed in vitro by T7 RNA polymerase (Promega, P2075) and biotin labeling mix (Roche, 11685597910). Then, MCF10CA1h cells were lysed by polysome lysis buffer (100 mM KCl, 5 mM MgCl$_2$, 10 mM HEPES, 0.5% Nonidet P-40, 100 U/mL recombinant RNase inhibitor, 400 μM ribonucleoside vanadyl complex, 1 mM DTT) for 15 min at 4 °C, and the supernatant was collected by centrifuging. A 3 μg Biotin-labeled RNA was added to 50 μL RNA structure buffer (10 mM Tris pH 7.0, 1 M KCl, 10 mM MgCl$_2$), then was denatured at 95 °C for 2 min, iced for 3 min and incubated at room temperature for 30 min step by step. Next, the RNA and the supernatant containing about 1 mg protein were added to 500 μL RIP buffer (150 mM KCl, 25 mM Tris pH 7.4, 0.5 mM DTT, 0.5% Nonidet P-40, 1 mM PMSF, protease inhibitors), and the complex was incubated for 2 h at room temperature. After 3–5 times of washing with DEPC water, streptavidin-packaged beads (Invitrogen, SA10004) were added to the complex for another 2 h incubation at room temperature. Finally, after washing beads 5 times by RIP buffer and 3 times by phosphate-buffered saline (PBS), proteins pulled down by RNA were eluted by SDS loading buffer for MS analyses or Western blotting.

**Cytoplasm/nucleus separation**. The PARIS$^{TM}$ Kit (Invitrogen, AM1921) was used for separating cytoplasmic/nuclear components of cells. Briefly, $10^6$–$10^7$ cells were resuspended by 300 μL pre-cooled cell fractionation buffer gently and were lysed on ice for 5 min. Then the supernatant obtained by centrifugation was the cytoplasmic component. The precipitate was washed one time by cell fractionation buffer, and the remaining precipitate was the nuclear component, which could be lysed by 100–200 μL cell disruption buffer. The cytoplasm and nuclear components can be used for extracting RNA or protein.

**Mouse experiments**. Female BALB/c nude mice and NOD/SCID mice aged 6–8 weeks were used in all animal experiments. For establishing the metastatic dormancy model in Fig. 1, nude mice were intravenously injected with $2 \times 10^5$ cells, and bioluminescence imaging data were collected by IndiGo v2.0.5.0. For the EdU labeling assay, nude mice were intravenously injected with $2 \times 10^5$ cells and were intraperitoneally injected with 100 ng/mouse EdU (ThermoFisher, C10640) 24 h before lung harvest. For analysis of the effect of *NAS1* on long-term metastasis, nude mice have intravenously injected with $5 \times 10^5$ CA1h-P1 cells or $2 \times 10^6$ CA1h-P2 cells, and NOD/SCID mice were orthotopically inoculated with $2 \times 10^5$ 4175-LM2 cells, for which the orthotopic tumors were surgically removed at the size of about 1 cm$^3$. For limiting dilution tumorigenesis assays, NOD/SCID mice were orthotopically injected with different numbers of cells as indicated in the figures. The depth of local invasion was measured from the tumor edges to the deepest points of invasive fronts after H&E staining of paraffin-embedded sections using the methods reported by previously studies[84,85]. To analyze circulating tumor cells, GFP-labeled breast tumor cells were orthotopically injected in mice. Mice were anesthetized when tumor volumes reached 0.5 cm$^3$ and 200 μL of blood was collected from the left ventricle. Genomic DNA was extracted from the blood samples with a DNA extraction kit (Tiangen, DP318-02) according to the manufacturer's protocol, followed by the qPCR analysis of the amount of *GFP* DNA in 100 ng genome DNA of each sample for relative quantitation of GFP + tumor cells. Investigators were not blinded to outcome assessment. All animal studies were approved by the Institutional Animal Care and Use Committee of Shanghai Institute of Nutrition and Health.

**Transcriptomic sequencing and GSEA analyses**. Transcriptomic sequencing was firstly performed in MCF10 breast cancer cell line series that include MCF10AT, MCF10CA1h, MCF10CA1a, CA1h-P1, and CA1h-P2. LncRNAs with expression fold changes >1.5 were identified by comparing MCF10CA1h with MCF10AT or MCF10CA1a and comparing CA1h-P2 with CA1h-P1, after excluding some lncRNAs with ultra-low abundance. Then the transcriptomes of CA1h-P2 with *NAS1* knockdown, MCF10CA1h, and MCF7 with *NR2F1* overexpression were sequenced to find the molecules affected by both *NAS1* and *NR2F1*. Genes with expression fold changes >1.5 were identified as the gene sets regulated by *NAS1* knockdown or *NR2F1* overexpression and further used for ssGSEA analysis. The genes with fold changes >3 were shown in the heatmap (Fig. 6a).

For GSEA, gene sets MCF7_ΔNp63_UP and MCF7_ΔNp63_DOWN were derived from the public expression profiles of MCF7 cells with ΔNp63 overexpression from the GEO database (GSE64953), and genes with expression fold changes >3 were selected. The signatures for epithelial and mesenchymal BCSCs (E-BCSC and M-BCSC) were derived from the data of Liu et al.[28] (GSE52262) by selecting genes with fold changes >4 in CD44$^+$/ALDH$^-$ cells versus non-CD44$^+$/ALDH$^-$ cells and in CD44$^-$/ALDH$^+$ cells versus non-CD44$^+$/ALDH$^-$ cells, respectively. The EMT signature was derived from the study of Mak et al.[72]. All the gene sets used for GSEA and ssGSEA are listed in Supplementary Table 4.

**Flow cytometry analyses**. For CD24/CD44 analyses, single-cell suspensions were incubated with the antibodies in recommended concentrations by the manufacturers' instructions at 4 °C for 30 min. After 3 times of PBS washing, cells were sorted by a MoFlo Astrios Flow Cytometer (Beckman, software: Summit 6.3.1) or analyzed by a Gallios Analyzer (Beckman, software: Gallios software 1.2). For ALDH activity analyses, the ALDEFLUOR Kit (StemCell, 1700) was used according

to the manufacturer's protocol. The specific ALDH inhibitor Diethylamino-benzaldehyde (DEAB) was used as a negative control. For cell cycle analyses, cells were fixed in pre-cooled 70% ethanol at 4 °C overnight. Then cells were washed with PBS and resuspended in PBS containing 100 μg/mL RNase A, following incubation at 37 °C for 30 min. Propidium iodide (PI) was then added to a final concentration of 20 μg/mL. After 15 min incubation at 4 °C, cells were analyzed by Gallios Analyzer. Flow cytometry data processing was performed with FlowJov10 (Tree Star, USA) and ModFit LT (Verity Software House, USA). The gating strategy of flow cytometric analyses was shown as Supplementary Fig. 12.

**Immunofluorescence (IF) staining**. Cells were seeded on coverslips in 24-well plates and were fixed with 4% paraformaldehyde for 15 min when growing to 60% confluence. 0.3% Triton X-100 containing PBS was used to permeate the cell membrane at room temperature for 10 min. The cover slides were then blocked in 3% BSA/PBS at room temperature for 1 h, and incubated with the primary antibody and fluorescent-labeled secondary antibody overnight at 4 °C and for 1 h at room temperature, respectively. Next, the nucleus was stained by DAPI (Roche, 10236276001) before mounting (Dako, S3023). Observation and photographing were performed with the confocal microscopy Cell Observer (ZEISS, Germany), and image processing and analysis were performed with Zen blue edition software (ZEISS, Germany). For tissue sections, lungs of mice were harvested at the designated time points with PBS lavage and fixed in 4% PFA for 1 h on ice, then were dehydrated by 30% sucrose PBS solution overnight and embedded in OCT (Sakura, 4583), followed by freezing at −80 °C. Tissues were sectioned to 10 μm thickness, and the IF staining and observation were performed as described above. For EdU staining, CLICK PLUS EdU 647 Imaging Kit (ThermoFisher, C10640) was used.

**Rapid amplification of cDNA ends (RACE) experiment**. The 5′/3′ RACE Kit, 2nd Generation (Roche, 03353621001) was used to amplify the 5′ and 3′ ends of *NAS1*. The experiments were carried out according to the manufacturer's instructions.

**Luciferase dual-reporter assay**. HeLa cells cultured in 48-well plates were transfected with the indicated firefly luciferase reporter plasmids, a Renilla luciferase plasmid, and the overexpression vectors or siRNAs. A 48 h later, the medium was discarded and cells were lysed with 60 μL luciferase lysis buffer (2 mM EDTA, 20 mM DTT, 10% glycerol, 1% Triton X-100 and 25 mM Tris-base, pH 7.8) for 1 h at room temperature. 10 μL lysate was added with 30 μL firefly luciferase assay buffer (25 mM glycylglycine, 15 mM potassium phosphate, 15 mM MgSO$_4$, 4 mM EGTA, 2 mM ATP, 10 mM DTT and 1 mM D-luciferin, pH 7.8) or 30 μL renilla luciferase assay buffer (0.5 M NaCl, 1 mM EDTA, 0.1 M potassium phosphate, 0.04% BSA and 2 μM coelenterazine, pH 7.4), then the luminescence was detected immediately by a Multimode Plate Reader (PerkinElmer, USA).

**Tumorsphere formation assay**. Totally, 5000 cells/well were seeded in 6-well ultra-low attachment plates (Corning, 3471) in 1:1 DMEM/F-12 supplemented with 1:50 B27 (Thermo Fisher Scientific, 12587010), 20 ng/mL epidermal growth factor (EGF, Thermo Fisher Scientific, PHG0311), 10 ng/mL basic fibroblast growth factor (bFGF, Sigma-Aldrich, F0291), 5 μg/mL heparin sulfate (Sigma-Aldrich, H3149), 5 μg/mL insulin (Roche, 11376497001) and 0.5 μg/mL hydrocortisone (Merck, 3867). For MCF7 cells, the heparin sulfate and hydrocortisone were not supplemented. Spheres with diameters larger than 100 μm were counted under the microscope after 2 weeks of culturing.

**Clinical analysis**. Breast cancer tissues for the correlation analysis of *NAS1* and NR2F1 were obtained from the clinical sample database of the Shanghai Institute of Nutrition and Health, Chinese Academy of Sciences. RNA and protein of these samples were extracted for *NAS1* and NR2F1 detection, respectively. NR2F1 and GAPDH protein bands by Western blotting were analyzed by software Image J (National Institutes of Health, USA) to calculate the relative expression level of NR2F1. For analyses of recurrence-free and metastasis-free survival of breast cancer patients, breast cancer tissues of 89 patients treated at Qilu Hospital of Shandong University were used for RNA extraction, and the expression level of *NAS1* was analyzed by qPCR. The effect of *NAS1* on recurrence-free survival was also analyzed by Kaplan–Meier Plotter clinical database[74]. Then, the RNA sequencing data of the TCGA breast cancer clinical cohort were used for correlation analyses of *NAS1* and EMT markers. Samples and prognostic information were obtained with informed patient consent and the approval from Research Review Boards of the Institute of Nutrition and Health and Qilu Hospital of Shandong University.

**Statistics and reproducibility**. The data presentation and statistical analyses are described in the figure legends. Data analyses were performed by GraphPad Prism 8 (GraphPad Software, USA). $P$ values < 0.05 were considered statistically significant. The experiments in vitro were repeated independently at least 3 times with similar results.

**Reporting summary**. Further information on research design is available in the Nature Research Reporting Summary linked to this article.

## Data availability

The RNA sequencing data generated in this study have been deposited in the Genome Sequence Archive (GSA) under accession code PRJCA006136 and National Omics Data Encyclopedia (NODE) under accession code OEP000853. The source data underlying Figs. 1a–f, 2a, c, d, f, g, i, j, 3f, h, i, k, 4c, f–I, 5b, d–g, i, k, m, n, 6a–c, f, h-I, 7a–g and Supplementary Figs. 1d–f, 2a, b, 3a, b, 4d–g, 5b, d–I, k–o, 6a, b, 7b, 8f–h, j–m, 9b-f, 10f–g are provided as a Source Data file. All the other data supporting the findings of this study are available within the article and its Supplementary information files and from the corresponding author upon reasonable request. A reporting summary for this article is available as a Supplementary Information file. Source data are provided with this paper.

Published online: 02 Septembe 2021

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

## Acknowledgements

We thank Zheng Yan, Xiang Miao, Lin Qiu, Yifan Bu, Yumei Wang, Kai Wang, Yujia Zhai, Yiting Yuan, Shuyang Yan and Jun Li at the Institute of Nutrition and Health core facilities for technical support. The study was funded by the Ministry of Science and Technology of China (2020YFA0112302, 2017YFA0103502), Chinese Academy of Sciences (QYZDB-SSW-SMC013), National Natural Science Foundation of China (81725017, 81872367, 82003090), and Postdoctoral Science Foundation of China (2020M671257).

## Author contributions

G.H. supervised this work. Y. Liu, P.Z., and G.H. designed the experiments and drafted the paper. Y. Liu, P.Z., Q.W., H.F., T.W., Y.W., Y.X., Y.H., P.T., M.C., C.M., and Y. Liang performed the experiments. L.Q., Qingcheng Y., and Qifeng Y. contributed in clinical sample collection and analysis. L.L. designed and supervised the mass spectrometry analysis. All authors discussed the results and commented on the paper.

## Competing interests

The authors declare no competing interests.
