## [Peer Review File · Nature Communications]

Reviewers' Comments:

Reviewer #1:

Remarks to the Author:

The revised manuscript successfully addressed most of the questions raised previously. Especially, the authors performed a series of experiments to demonstrate the role of endogenous NR2F1 downstream of NR2F1-AS1, which is well done.

While the study focuses on the role of NR2F1-AS1 in promoting invasion, it is clear from the newly added Fig. S5B that NR2F1-AS1 has a major role in suppressing cell proliferation not just in the lung, but also in primary tumors. This anti-proliferative role of NR2F1-AS1 could be one of the reasons that high NR2F1-AS1 level is correlated with good prognosis. Therefore, the anti-proliferative role of NR2F1-AS1 needs to be further clarified in the discussion in the context of tumor development and patient prognosis since its correlation with good prognosis could be more related to its role in suppressing proliferation than its role in promoting invasion.

Reviewer #3:

Remarks to the Author:

My previous concerns were adequately addressed, and the manuscript is novel and innovative.

Reviewer #4:

Remarks to the Author:

In this manuscript, the authors investigated the functions of lncRNA NAS1 in controlling tumor stem cell dormancy. They found NAS1 is highly expressed in mesenchymal, invasive stem cell subpopulations, and helps to maintain the dormant state of disseminated tumor cells. Mechanistically, NAS1 recruits PTBP1 to the 5'UTR region of NR2F1 mRNA and promotes NR2F1 translation, which in turn inhibits Δ Np63/miR205 transcription and results in cell dormancy and reduced tumorigenicity. In general, this MS delivered an interesting story and clarifies a lncRNA-centered regulatory mechanism of tumor cell dormancy. Since this MS had been revised according to the previous reviewers' comments and provided several additional data panels and literature review, the data was enough to support the conclusion. There are only some minor issues need to be addressed.

1. In figure 1B&C, 2F, the representative images couldn't reflect the percentage of GFP+ cells and EDU+ cells as shown in the statistical column. It seems MCF10CA1a/P1 inoculation mice had more GFP+ cells, and EDU+ cells didn't significantly increase than that in MCF10CA1h. Also, the authors should count the percentage of GFP+ EDU+ cells instead of EDU+ cells to show the active proliferative metastatic cells.
2. In figure 3B&C, the authors should detect the percentage of CD24-/CD44+ cells, instead of CD44+ alone.
3. Page 3 Line 11, "recozenized" should be "recognized".

Reviewer #5:

Remarks to the Author:

Overall, the authors have addressed the previous concerns.

REVIEWERS' COMMENTS

Reviewer #1 (Remarks to the Author):

The revised manuscript successfully addressed most of the questions raised previously. Especially, the authors performed a series of experiments to demonstrate the role of endogenous NR2F1 downstream of NR2F1-AS1, which is well done.

Thanks for your recognition of our work. Your suggestions are very helpful for us to improve the manuscript.

While the study focuses on the role of NR2F1-AS1 in promoting invasion, it is clear from the newly added Fig. S5B that NR2F1-AS1 has a major role in suppressing cell proliferation not just in the lung, but also in primary tumors. This anti-proliferative role of NR2F1-AS1 could be one of the reasons that high NR2F1-AS1 level is correlated with good prognosis. Therefore, the anti-proliferative role of NR2F1-AS1 needs to be further clarified in the discussion in the context of tumor development and patient prognosis since its correlation with good prognosis could be more related to its role in suppressing proliferation than its role in promoting invasion.

Thanks for the suggestion. Actually we totally agree with the reviewer at this point. The anti-proliferation and pro-invasion roles of *NR2F1-AS1* seem contradictory towards tumor progression, but both contribute to the accumulation of dormant disseminated tumor cells in metastatic organs. It is the anti-proliferation role, which suppresses primary tumor growth and enhances dormancy of disseminated tumor cells, that likely contributes to the good prognosis of patients with higher *NR2F1-AS1* expression, while its pro-invasion role, although promoting tumor dissemination, might not be sufficient to reverse the good prognostic effect as the disseminated tumor cells fail to give rise to metastases. We have updated the Discussion section to clarify this point in the revision.

Reviewer #3 (Remarks to the Author):

My previous concerns were adequately addressed, and the manuscript is novel and innovative.

We thank the reviewer for her/his help to improve our manuscript.

Reviewer #4 (Remarks to the Author):

In this manuscript, the authors investigated the functions of lncRNA NAS1 in controlling tumor stem cell dormancy. They found NAS1 is highly expressed in mesenchymal, invasive stem cell subpopulations, and helps to maintain the dormant state of disseminated tumor cells. Mechanistically, NAS1 recruits PTBP1 to the 5' UTR region of NR2F1 mRNA and promotes NR2F1 translation, which in turn inhibits Δ Np63/miR205 transcription and results in cell dormancy and reduced tumorigenicity.

In general, this MS delivered an interesting story and clarifies a lncRNA-centered regulatory mechanism of tumor cell dormancy. Since this MS had been revised according to the previous reviewers' comments and provided several additional data panels and literature review, the data was enough to support the conclusion. There are only some minor issues need to be addressed.

1. In figure 1B&C, 2F, the representative images couldn't reflect the percentage of GFP⁺ cells and EDU⁺ cells as shown in the statistical column. It seems MCF10CA1a/P1 inoculation mice had more GFP⁺ cells, and EDU⁺ cells didn't significantly increase than that in MCF10CA1h. Also, the authors should count the percentage of GFP⁺ EDU⁺ cells instead of EDU⁺ cells to show the active proliferative metastatic cells.

This is an oversight of our data presentation. As the statistics showed, lung sections of mice inoculated with MCF10CA1h and P2 cells had more GFP⁺ cells but a lower proportion of EdU⁺ cells in GFP⁺ tumor cells than those with MCF10CA1a and P1 cells in Fig. 1b&c. Similar results were observed in the mice inoculated with NAS1-overexpressing cells in Fig. 2f. The dye indicating text and scale bars on the images masked some GFP⁺ cells, and the Y-axis labeling ("EdU⁺ tumor cells %") in the last submission was also confusing. In this revision, we have updated the

representative images with the more illustrative ones, moved the dye indicating text and scale bars on the images, and changed the Y-axis labeling (“% EdU+ tumor cells in GFP+ cells”) to avoid the ambiguity.

2. In figure 3B&C, the authors should detect the percentage of CD24-/CD44+ cells, instead of CD44+ alone.

Thanks for your suggestion. We reanalyzed these cells with CD24 and CD44 antibodies, and new images are shown in Fig. 3b&c. In Fig. 3b, since the whole CA1h-P2 cell population was CD24^{low/-}, we only need to use a differentiating line of CD44 staining to analyze the difference of CD24⁻CD44⁺ percentage caused by *NASI* knockdown.

3. Page 3 Line 11, “recogenized” should be “recognized”.

It was our fault. We corrected the spelling.

Reviewer #5 (Remarks to the Author):

Overall, the authors have addressed the previous concerns.

We thank the reviewer for her/his help to improve our manuscript.